# Plasma Exchange versus Intravenous Immunoglobulin in Worsening Myasthenia Gravis: A Systematic Review and Meta-Analysis with Special Attention to Faster Relapse Control

**DOI:** 10.3390/biomedicines11123180

**Published:** 2023-11-29

**Authors:** Mark Pavlekovics, Marie Anne Engh, Katalin Lugosi, Laszlo Szabo, Peter Hegyi, Tamas Terebessy, Gabor Csukly, Zsolt Molnar, Zsolt Illes, Gabor Lovas

**Affiliations:** 1Center for Translational Medicine, Semmelweis University, Üllöi út 26, 1085 Budapest, Hungary; mrkpavlekovics@gmail.com (M.P.);; 2Department of Neurology, Jahn Ferenc Teaching Hospital, Köves út 1, 1204 Budapest, Hungary; 3Department of Neurology, Bajcsy-Zsilinszky Hospital, Maglódi út 89–91, 1106 Budapest, Hungary; 4Institute of Pancreatic Diseases, Semmelweis University, Baross utca 22–24, 1085 Budapest, Hungary; 5Institute for Translational Medicine, Medical School, University of Pécs, 12 Szigeti Street, 7623 Pécs, Hungary; 6Department of Orthopedics, Semmelweis University, Üllői út 78/b, 1082 Budapest, Hungary; 7Department of Psychiatry and Psychotherapy, Semmelweis University, Balassa utca 6, 1083 Budapest, Hungary; 8Department of Anesthesiology and Intensive Therapy, Semmelweis University, 78 Üllöi St, 1085 Budapest, Hungary; 9Department of Neurology, Odense University Hospital, Winslows Vej 4, 5000 Odense, Denmark; 10Institute of Clinical Research, University of Southern Denmark, Campusvej 55, 5230 Odense, Denmark

**Keywords:** myasthenia gravis, relapse, plasma exchange, intravenous immunoglobulin, meta-analysis

## Abstract

Currently used rescue interventions to prevent rapid myasthenic deterioration are plasma exchange (PLEX) and intravenous immunoglobulin (IVIG). We investigated the evidence to determine whether the two methods were interchangeable or whether one was superior to the other. This review was registered on PROSPERO (CRD42021285985). Only randomized controlled trials (RCTs) comparing the efficacy and safety of PLEX and IVIG in patients with moderate-to-severe myasthenia gravis (MG) were included. Five major databases were systematically searched (PubMed, CENTRAL, Embase, Scopus, and Web of Science). Odds ratios (OR) with 95% confidence intervals (CI) were calculated for adverse events and mean differences (MD) for changes in quantitative myasthenia gravis scores (QMG). Three RCTs met the inclusion criteria. Two investigating 114 patients in total were eligible for meta-analysis to analyze efficacy and safety. For the change in QMG score, the MD was −2.8 (95% CI: −5.614–0.113), with PLEX performing better. For adverse events, an OR of 1.04 was found (95% CI: 0.25–4.27). This study demonstrated a low risk of bias in evaluating treatment efficacy but indicated a high risk of bias in assessing procedural safety outcomes. Although the results did not show any significant difference, there was a tendency indicating faster efficacy of PLEX in the first two weeks of treatment. In such a critical clinical condition, this tendency may be clinically meaningful, but further studies should clarify this benefit.

## 1. Introduction

Myasthenia gravis (MG) is an autoimmune disease caused by pathogenic autoantibodies binding to variable parts of the neuromuscular junction receptor complex, resulting in muscle weakness and fatigue [1,2]. The annual incidence is between 1.7 and 21.3/1,000,000, with separate peaks in the third and sixth [3,4] decades. Although the prognosis of MG is improving, the overall survival of patients is still shorter than that of the general population [4,5].

Myasthenic respiratory crisis is the most feared life-threatening complication of MG; it requires intensive care unit (ICU) admission [6,7]. It is most often triggered by infection, surgery, or the postpartum period [8]. Thymic abnormalities, such as follicular hyperplasia or the presence of thymoma, predict a worse prognosis [9]. Approximately 15–20% of patients experience respiratory crises at least once in their lives [10,11]. The mortality rate of the first crisis is approximately 2–5% [10]. The rapid detection of the relapse is crucial because it requires fast intervention. The currently used rescue interventions to prevent progression into crisis are intravenous immunoglobulin (IVIG) and plasma exchange (PLEX).

PLEX has been used worldwide to treat MG since 1976 [12]. It eliminates the pathogenic autoantibodies, immune complexes, and cytokines from the plasma. PLEX requires special instruments and trained and experienced personnel. Due to the nature of plasma exchange interventions, it is necessary to insert intravenous lines into more durable veins. This is not the case with intravenous immunoglobulin treatment, which is a plain infusion (Table 1). In addition, obtaining and maintaining suitable venous access can present challenges (e.g., central vein cannulation), with further consideration of potential complications (e.g., central vein thrombosis). Although invasive, PLEX is safe and effective [13]. The European Federation of Neurological Societies [14] and the MG Foundation of America [11] consensus guidelines [15] recommend PLEX in myasthenic relapse. Despite the unquestionable role of PLEX, only a limited number of well-designed studies are available to determine exactly how useful it is in MG exacerbations [13,16].

IVIG preparations contain human immunoglobulin extracted from pools of blood from different donors. The exact mode of action is unclear. It appears to involve both T and B cell regulatory processes, the embryonic Fc receptor, and complement down-regulation [17]. IVIG has been used for a decade as a rescue therapy in MG [17,18,19,20]. Despite its price and availability, it is widely used in MG centers. It is less troublesome for patients and is more feasible to apply in an outpatient setting. Placebo-controlled randomized studies provide level-one evidence indicating that IVIG is effective and safe in the treatment of an MG relapse [18,21]. Notably, IVIG supply [22] shortages are prevalent, particularly in economically less developed countries. The production of IVIG relies on a continuous supply of voluntary blood donors, and its utilization spans various medical fields. Furthermore, the manufacturing process is costly.

Table 1 provides a concise summary of the advantages and disadvantages of PLEX and IVIG in myasthenia gravis treatment. PLEX shows fewer advantages, while IVIG faces limited availability and relies on blood donors, contributing to supply challenges.

According to consensus guidelines and recent meta-analyses, either PLEX or IVIG can be used in cases of rapid deterioration [18,21]. A few randomized controlled studies [18,21] suggest that the two interventions are equal in efficacy, but there is no clear evidence of interchangeability or superiority.

The aim of this study was to carry out a systematic review of the existing evidence from randomized controlled trials comparing the efficacy and safety of PLEX versus IVIG in the management of moderate to severe MG, with special attention to faster relapse control.

In the evolving landscape of myasthenia gravis management, the introduction of novel therapeutic approaches signifies a transformative shift. This paper does not aim to provide a detailed review of the new treatment possibilities. Due to their fast therapeutic effects, it is crucial to highlight anti-FcRn antibody therapy and complement inhibitor therapies as potential future options for managing the MG crisis.

## 2. Methods

The systematic literature review and meta-analysis followed the recommendations of the PRISMA 2020 guidelines (Appendix A) and the Cochrane Handbook. The protocol of this study was registered in PROSPERO (identifier: CRD42021285985). A systematic search was performed in five major databases using predefined search terminology one day after PROSPERO registration.

### 2.1. Clinical Question

What is the efficacy and safety profile of PLEX compared to IVIG in myasthenia gravis relapse?

### 2.2. Hypothesis

Considering PLEXs direct elimination of autoantibodies and IVIGs gradual immune response modulation, we hypothesize that PLEX is superior to IVIG in terms of efficacy for moderate-to-severe myasthenia gravis patients.

### 2.3. Eligibility Criteria

Studies comparing IVIG vs. PLEX as treatment for patients with moderate to severe MG (QMG ≥ 10) were eligible for inclusion. Studies were included in our analysis if they provided data on QMG and adverse events. Only randomized controlled trials were eligible for inclusion. Exclusion criteria were as follows: studies that did not fulfill the requirements of study design (non-RCT) or meet our PICO (Figure 1).

### 2.4. Information Resources

Five major databases were used for systematic search: MEDLINE (using PubMed), the Cochrane Central Register of Controlled Trials (CENTRAL), Web of Science, Scopus, and Embase. PROSPERO registration was completed on 22 October 2021, the search was performed afterwards. No filter was applied, and no restrictions were made based on the language or publication date of the articles.

### 2.5. Search Strategy

During the systematic search, the following search key was used: (((immunoglobulin OR “immune globulin” OR autoantibody) AND (intravenous OR iv)) OR plasmapheresis OR “plasma exchange” OR plex) AND “myasthenia gravis” AND random*.

### 2.6. Selection Process

The selection was performed by two independent review authors (KL and MP) after duplicates were removed, in a two-stage process—first by title and abstract; then by full text. Disagreements were resolved by consensus following the discussion. The degree of agreement was quantified with Cohen’s kappa statistics at each stage of selection [23].

### 2.7. Data Collection Process

From the eligible full texts, data were collected by one of the authors (PM) using Microsoft Excel (2023) 16.72 version^®^, then double-checked by a second author.

### 2.8. Data Items

The Quantitative Myasthenia Gravis Score (QMG) was used to evaluate effectiveness [24,25]. The QMG contains no subjective items and assesses 13 major symptoms on a scale of 0–3. A higher score indicates a more severe condition. The maximum score for QMG is 39. According to overall severity, three groups were created: mild (0–9), moderate (10–16), and severe (16–39) myasthenia. QMG was chosen as an outcome measure based on the recommendation of the MGFA working group, as it can be used to determine the severity of the symptoms objectively [26]. The use of other scales was not an exclusion.

The safety of the therapy was assessed by the number of adverse events reported during treatment. The number of adverse events is a frequently reported piece of information that provides an opportunity for objective comparison.

The data were extracted from the standardized data collection sheet. The following information was collected: study name, first author, publication year, Digital Object Identifier (DOI), language, contact details, study design, study duration, randomization, blinding, country, number of participating centers, total number of participants, number of patients who were randomized and the number of those who completed the study, age, gender, description of enrolled patients, concomitant therapy in both treatment arms, method and frequency of administration, total duration of treatment, number of patients receiving (randomized and completed), patients who underwent thymectomy, and all existing data on the investigated outcomes were collected. Data items were previously recorded in PROSPERO.

### 2.9. Study Risk of Bias Assessment and Grading

Two independent authors have conducted an assessment of the risk of bias with the Cochrane Risk of Bias Tool for Randomized Trials version 2 (RoB 2 [27]). All disagreements were resoluted by consensus. To assess the level of evidence, GRADE was used, but due to the limited number of trials, this was not interpretable.

### 2.10. Synthesis Methods

For calculations and graphs, *meta* [28] and *dmetar* [29] software packages were used in the R programing environment.

Odds ratios (OR) with a 95% CI were used as effect measures for dichotomous outcomes. ORs were calculated after extracting the total number of patients in each group and the total number of patients with the event from each study. The raw data from the selected studies were pooled with a random-effects model applying the Mantel–Haenszel method [30,31,32].

In the case of the pooled results, the exact Mantel–Haenszel method (without continuity correction) was applied to handle the zero cell count. For individual studies, the problem of zero cell count was corrected by continuity correction of the treatment arm.

Both the Paule–Mandel method [33] and the Q-profile method [34] were used to calculate the confidence interval of tau^2^ for estimating tau^2^. We assessed statistical heterogeneity between studies using the Cochrane Q-test and i2 values.

To summarize the results, Forest Plots and Drapery Plots were used. Where appropriate, the prediction intervals of the results (i.e., the range of expected effects of future studies) were presented as recommended by IntHout et al. [35].

To assess publication bias, a funnel plot of the logarithm of the effect size and a comparison with the standard error of each study were used. The Egger test was used to assess publication bias, and the Harbord method was used to estimate the test statistic (see Appendix A).

## 3. Results

### 3.1. Search and Selection

A detailed summary of the selection process is found in the included PRISMA flowchart (Figure 2). Altogether, 3.769 publications were identified in the five databases. During the title and abstract selection, we found 14 eligible full texts, but only three of them were eligible for data extraction. One study [36] was excluded from quantitative analysis because the measurements used were not comparable with those of other studies. The RCT published by Gajdos [37] on this topic did not meet the PICO framework criteria and was, therefore, not included in the meta-analysis section.

Preferred Reporting Items for Systematic reviews and Meta-Analyses (PRISMA 2020) flow diagrams are shown in Figure 2. Due to the rigorous criteria, only two studies could be included in the meta-analysis. The rest of the studies had to be excluded as the examined parameters did not sufficiently overlap (e.g., studies examining the overall effect of thymectomy, or using scoring systems other than QMG).

### 3.2. Basic Characteristics of Included Studies

The baseline characteristics of the enrolled studies are described in Table 2.

### 3.3. Efficacy Parameters

Two studies were included in the meta-analysis for outcome, covering a total of 114 patients. The mean difference (95% confidence interval) of points in the QMG between the groups was −2.8 (−5.614–0.113) (*p* = 0.0596) (Figure 3). The result of the i2 test was 0.652 (95% CI: 0–0.921), and the variance of true effects (tau^2^) was 2.93 (SD: 1.71).

### 3.4. Adverse Events

A total of 2 studies had satisfied the stringent criteria for analysis, covering a total of 114 patients, of whom 49 had experienced an adverse event (Figure 4). The pooled odds ratio was 1.04 (95% CI: 0.25–4.27) (*p* = 0.9591). Heterogeneity between studies, expressed as i2 was 0.38. The value of random effects tau^2^ was 0.51.

### 3.5. Risk of Bias Assessment

As the study number is limited (<10), the results should be handled with caution. The results of the risk of bias assessment are presented in (Table 3).

The Risk of Bias Assessment in Table 3 demonstrates that the Efficacy Related Parameters fall under the low risk of bias in both studies; however, the Adverse Event Related Parameters did not satisfy the low risk of bias criteria.

### 3.6. Publication Bias and Heterogeneity

Funnel plots were generated; however, due to the limited number of studies, they are not interpretable. The funnel plots are found in the Appendix A.

## 4. Discussion

This study focuses on the comparison of rescue treatments (PLEX and IVIG) for moderate (QMG score 10–16 points) to severe (QMG score above 16 points) myasthenia gravis. The results of all four suitable randomized controlled trials currently available on this topic were analyzed. All of these studies compared PLEX and IVIG [36,37,38,39] and came to the conclusion that PLEX and IVIG were equally effective in the long run.

We only included studies in which the effectiveness of treatments was measured by changes in the Quantitative Myasthenia Gravis Score. In designing this study, we have sought to select an objective test scale, but we have also conducted a literature review to evaluate other scales. We identified three randomized controlled trials that met our criteria; however, one was not included in the meta-analysis as it used a different scale [37]. The clinical trial reported by Ronager was also excluded from the meta-analysis, as he carried out a controlled cross-over design including 12 patients [36]. It is applied to the Osserman classification to assess disease severity, and a modified QMG was used to assess treatment efficacy. He noted that the clinical effect of plasma exchange occurred earlier [36].

Although equally effective in the long run, it remains to be determined in a clinical situation with a rapidly deteriorating patient which intervention brings more rapid improvement. Comparing the two interventions, PLEX and IVIG, those who received PLEX treatment had on average 2.8 points lower scores after two weeks of treatment than those in the IVIG treatment group. Albeit not significant, this result is in accordance with the above-mentioned observation of Ronager [36] and the study by Liu [39].

Qureshi, in a multicenter retrospective study, looked at the treatment of 54 patients with myasthenic [21] crises. They also found that PLEX was more effective after two weeks. Guptill [40] studied 110 MuSK+ patients. He concluded that PLEX was the preferred treatment for rapid worsening. Gajdos [37] and Barth [38] did not find any differences between the treatments during their randomized controlled study; however, the small number of cases limited the interpretation of the study by Gajdos. Murthy also found in a large retrospective study [41] that the two interventions were equally effective in myasthenic crises. The two largest systematic reviews by Ortiz-Salas [42] and Gajdos [16] did not find any difference in efficacy.

Considering a maximum of three points per symptom can be assigned in the QMG, the data revealing an average reduction of 2.8 points after two weeks of PLEX treatment compared to the IVIG group suggests a potential clinical benefit. This result indicates that, on average, one symptom is alleviated at the end of the two-week treatment period using PLEX compared to IVIG. However, it is essential to note that these findings, while suggestive, did not achieve statistical significance, likely due to the study’s limited sample size.

In addressing the potential severity of MG, achieving rapid improvement is indispensable. The fast therapeutic impact of PLEX assumes paramount importance, especially in respiratory crises; therefore, further studies should clarify its benefit over IVIG in respiratory crises. Our study further supports the faster effect of PLEX by investigating a larger number of cases.

As a secondary outcome, our study aimed to evaluate the safety of PLEX and IVIG.

The safety of a given therapy might be represented by the total number of reported adverse events during treatment (Table 4). According to Ronager [36], there are generally more frequent but fewer serious adverse events with IVIG. Liu [39] did not describe any serious adverse events with either treatment, but fewer events were recorded with IVIG. It has to be noted that the definition of adverse events differed in these studies, and certain events were assessed with different levels of stringency.

On the basis of our analysis of the pooled data of 114 patients, there was no safety difference between PLEX and IVIG.

Our results show no statistical difference regarding AE occurrence and fall into the high risk of bias category. A notable disparity is evident in the reported number of adverse events, with Barth documenting a total of forty-five adverse events, whereas Liu reported only three. This suggests differences in the detection or reporting of adverse events between the two studies.

Table 4 compiles the reported incidence of adverse events in the studies, highlighting various side effects associated with both interventions. Severe events, such as myocardial infarction, were observed in the PLEX group, while occurrences of hemolytic anemia during IVIG treatment were noted, albeit in a limited number of included patients.

Adding nuance to the analysis, an examination of the randomized controlled study conducted by Ronager [36] reveals a total of 21 reported adverse events. Specifically, during plasma exchange, 7 events (affecting 5 patients) were documented, whereas during IVIG, 14 events (involving 9 patients) were reported.

Summarizing the findings from three randomized controlled trials, it can be concluded that the invasiveness associated with PLEX does not correlate with a significantly higher number of adverse events.

Our study highlights that not only the counts of AEs but their severity in trials should be uniformly reported as well. If only the number of side effects is evaluated, one cannot distinguish between mild and serious side effects. In our assessment, common yet manageable adverse effects hold comparatively less significance than those necessitating the discontinuation of initiated interventions. Such interruptions impede the timely realization of therapeutic efficacy or, alternatively, may herald a deterioration in the patient’s clinical state.

It remains a subjective clinical question whether a treatment with a fewer, but at the same time more severe, side effects profile should be preferred.

Data from registries may provide new insights on PLEX vs. IVIG in real-world settings.

Despite the well-established understanding that different serostatus profiles in MG exhibit distinct clinical presentations and varied therapeutic responses, randomized investigations comparing the responses to PLEX and IVIG remain lacking.

According to previous findings, MuSK+ patients display a more moderate positive response to steroid therapy, coupled with increased side effects and diminished efficacy with pyridostigmine treatment [43]. IVIG therapy yields responses ranging from 11% to 61%, suggesting a preference for PLEX in MuSK+ patients, a distinction not observed in the literature concerning acetylcholine receptor-positive cases [43,44,45].

The randomized controlled trials we reviewed predominantly involved AChR+ patients; Barth’s study incorporated 56 AChR+ cases alongside 4 MuSK+ and 17 seronegative patients. Subgroup analysis from existing data remains inconclusive.

An additional unresolved aspect is whether AChR titers correlate with disease severity and provide informative data for assessing therapy efficacy. While numerous studies in this field consistently conclude that there is no reliable correlation between changes in individual AchR antibody concentrations and the severity of clinical symptoms in patients with MG [46], other investigations present a divergent perspective [47]. Liu observed a noteworthy reduction in Titin and AChR antibody values, highlighting a more pronounced decrease in Titin antibodies within the PLEX group compared to the IVIG group [48]. Correlation analysis revealed a longitudinal association between the reduction of Titin antibodies and the decline in QMG scores. Additionally, Barth noted a reduction in AChR antibodies with treatment, although no significant differences were observed between groups [38]. Barth’s analysis of baseline covariates underscored that disease severity, QMGS, and seropositivity predicted a superior response to therapy, emphasizing that patients with more severe disease and the presence of antibodies exhibited a more favorable treatment response. Baseline factors such as age, gender, duration of MG, thymoma status, thymectomy status, and electrodiagnostic status did not predict the response to therapy. This comprehensive overview illuminates the intricate nuances surrounding serostatus-specific responses to PLEX and IVIG in MG, emphasizing the imperative need for further investigations to refine therapeutic strategies.

## 5. Conclusions

Although publications in medical literature indicate that PLEX and IVIG are equally effective in the long run [11,21,24], it is surprising to see that the number of properly designed RCTs comparing PLEX with IVIG in MG relapse is very limited: we found only four RCTs available on this topic, and only two were suitable for strict analysis.

Albeit statistically not significant, PLEX may have a faster beneficial effect on myasthenia crisis prevention, with a difference of more than 2.5 points in the improvement of the QMG score at week two when compared to IVIG therapy.

No significant difference was found between the two modalities in terms of the side effect profile. The high risk of bias should be taken into account when assessing this result.

Further studies would be appreciated to address this topic with homogeneous disease groups, a uniform treatment regimen, and adequate follow-up time.

### 5.1. Strengths and Limitations

Regarding the strengths of our analysis, we followed our protocol (id: CRD42021285985), which was registered in advance. Rigorous methodology was applied. To the best of our knowledge, this is the first meta-analysis of randomized, controlled studies. The study highlights the shortcomings of the evidence on the subject. The review’s further strength lies in its focused exploration of a specific topic, clearly underscoring that our understanding of the effectiveness and equivalence of PLEX and IVIG is grounded in only two randomized controlled trials.

Concerning the limitations of this study, it is crucial to highlight that the analysis relied on data from only two trials. Another constraint is the variation in therapeutic regimens and dosages, with different amounts of PLEX treatment and varied IVIG dosages measured in g/kg. The mild heterogeneity within the evaluated population represents a moderate limitation. Additionally, the presence of a moderate and high risk of bias in certain domains introduces an additional limitation.

### 5.2. Implications

#### 5.2.1. Implications for Research

Further data collection is needed to assess the issue more accurately. Well-designed observational clinical trials or an international MG registry might give additional insight into this issue.

When a randomized controlled trial is designed, consideration should be given to a heterogeneous group of patients, with particular attention to serotype and age of onset. When selecting the patient group, the number of crises, previous response to PLEX/IVIG, and immunosuppressive therapy have to be taken into account. It is recommended that the number of PLEX and IVIG treatments and the dosage be planned to follow the recommendation of MGFA [24]. In terms of outcomes, the widespread use of the more objective measure of QMG would be preferable [49]. The problem with subjective scales like MG-ADL or other patient-reported outcome measures is the inherent bias that objective scales are completely free of. Weighting adverse events, whether they required intervention or persisted longer than a given time, would be clinically important. It would also be preferable to record the length of hospitalization and changes in therapy.

#### 5.2.2. Implications for Practice

The use of scientific results in patient care is a fundamental aim of our research. Despite the limited data available, the following conclusions are worth considering by clinicians for practical use: PLEX and IVIG are equally effective in MG treatment in the long run.

As we analyzed exclusively randomized trials with a sufficient number of cases, it might only be suggested that PLEX might provide faster relapse control in an acute rescue situation. This might come into consideration when both treatment options are available.

Furthermore, advocating for the prioritization of PLEX is grounded in the understanding that the combination of PLEX and IVIG yields benefits only when PLEX precedes IVIG or new therapeutic options, like monoclonal antibodies. In such instances, a synergistic effect can be attained, as opposed to the scenario where the impact of IVIg is nullified by the elimination of all IgG through PLEX. This rationale provides a compelling justification for initiating PLEX first, particularly in the most severe cases.

As a prospective suggestion, it is recommended to monitor the role of monoclonal antibodies, with a particular emphasis on FcRn receptor inhibitors (efgartigimod) and complement therapies (eculizumab) in the context of myasthenic crises [50,51,52,53]. Given that the effects of these therapies can manifest rapidly, potentially within a week, and precisely target specific immunopathological pathways, they may play a pivotal role with fewer side effects in future crisis management. However, official studies are necessary to determine whether they are comparable to standard treatments, such as IVIG or PLEX, in achieving significant and rapid clinical improvement. It is important to note that the mentioned therapies are particularly expensive, imposing a substantial burden on the insurance system, and accessibility can be challenging in some countries. Therefore, PLEX and IVIG are likely to remain prominent players in everyday practice for an extended period of time.

## Figures and Tables

**Figure 1 biomedicines-11-03180-f001:**
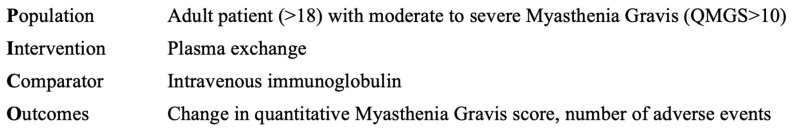
It explains the PICO criteria for the meta-analysis. This figure illustrates the studied population, the intervention, and the comparator, as well as presenting the outcomes. In our study, adult patients with moderate or severe myasthenia gravis were investigated in the adult population.

**Figure 2 biomedicines-11-03180-f002:**
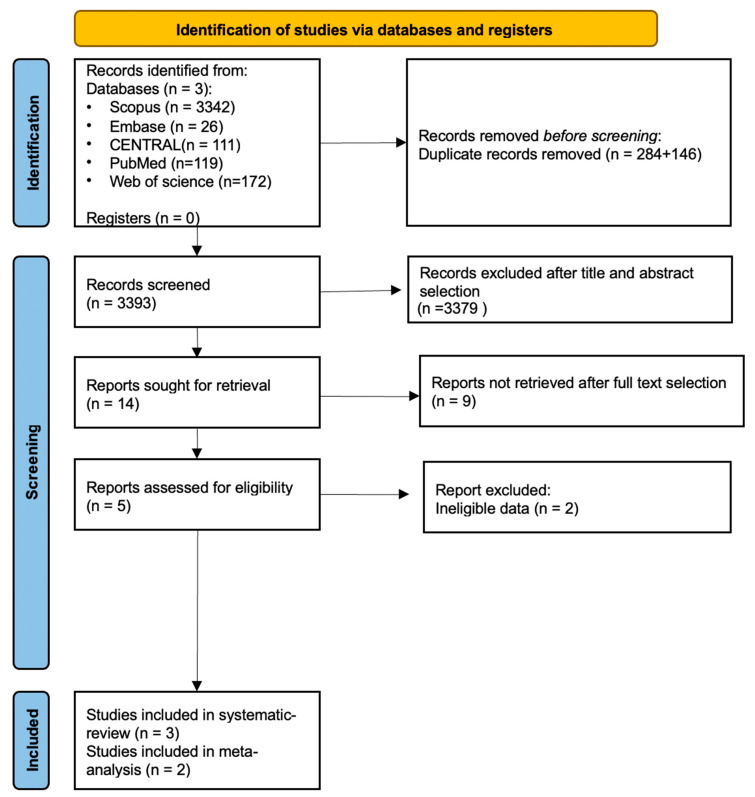
Preferred Reporting Items for Systematic reviews and Meta-Analyses (PRISMA 2020) flow diagrams are shown in the Figure. Due to the rigorous criteria only two studies could be included in the meta-analysis. The rest of the studies had to be excluded as the examined parameters did not sufficiently overlap (e.g., studies examining the overall effect of thymectomy, or using scoring systems other than QMG).

**Figure 3 biomedicines-11-03180-f003:**
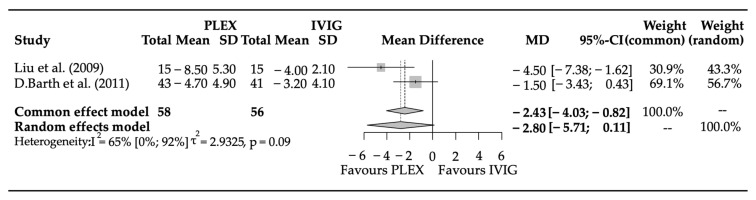
Meta-analysis results comparing the effectiveness of PLEX and IVIG in severe to moderate MG patients were measured by QMG [38,39]. Within the examined timeframe of two weeks, PLEX might show better efficacy in patients with deteriorating myasthenic symptoms, although the difference may have missed the level of significance. In the case of rapidly worsening myasthenic symptoms, this difference deserves attention by clinicians; nevertheless, its confirmation requires further clinical studies. This tendency is also demonstrated by the Common Effects vs. Random Effects model comparison.

**Figure 4 biomedicines-11-03180-f004:**
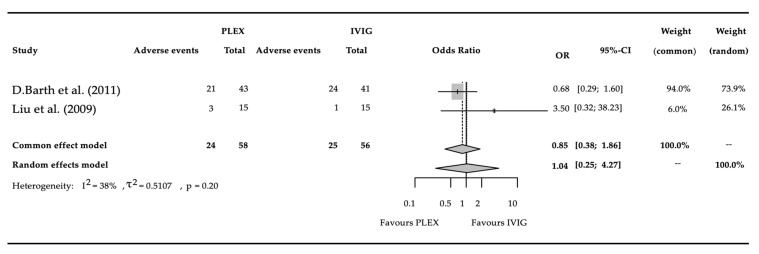
Meta-analysis results comparing the number of adverse events within 30 days between PLEX and IVIG treatments in severe to moderate MG [38,39]. The graph clearly shows no major difference in the overall AE occurrence between the two interventions. It is important to note that the results had a high risk of bias (see Table 2).

**Table 1 biomedicines-11-03180-t001:** It provides a concise summary of the advantages and disadvantages of PLEX and IVIG in myasthenia gravis treatment. PLEX shows fewer advantages, while IVIG faces limited availability and relies on blood donors, contributing to supply challenges.

Aspect	Plasma Exchange	Intravenous Immunoglobulin
**Advantages**	Suitable for immediate intervention Rapid removal of pathogenic antibodies	Suitable for immediate intervention Requires minimally invasive intervention Suitable for outpatient setting Does not remove the applied therapy
**Disadvantages**	Invasive procedure requiring vascular access Requires special instruments, and trained personnel Removed the applied therapy	Expensive treatment option Shortage of supply may occur

**Table 2 biomedicines-11-03180-t002:** It presents the baseline characteristics of the examined trials. The table demonstrates that the two included studies were single-center, randomized, evaluator-masked trials. Additionally, it illustrates that the applied interventions were carried out based on different protocols. (PLEX: plasma exchange; IVIG: intravenous immunoglobulin).

	Country	Number of Centers	Randomized	Blinded	Mean Age of PLEX Group	Number of Patients/ Sexes in PLEX Group	Baseline QMG of PLEX Group	Number of PLEX Treatments	Mean Age of IVIG Group	Number of Patients/ Sexes in IVIG Group	Baseline QMG of IVIG Group	IVIG Dosage
Barth, D. et al. (2011) [38]	Canada	Single-centered	Yes	Evaluator masked	58 ± 17	17M/24F	14.44 ± 3.8	5/every second day	57 ± 18	19M/24F	14.26 ± 4.0	2/2 day (1g/kg/d)
Liu et al. (2009) [39]	China	Single-centered	Yes	Evaluator masked	55.2 ± 1.4	9M/6F	19.4 ± 2.2	3/24–48 h	53.2 ± 1.7	8M/7F	16.5 ± 1.7	5/5day (0.4 g/kg/d)

**Table 3 biomedicines-11-03180-t003:** The Risk of Bias Assessment in Table 3 demonstrates that the Efficacy Related parameters fall into low risk of bias in both studies; however, the adverse event related parameters did not satisfy the low risk of bias criteria.

Efficacy related parameters							**D1:** Randomization process **D2:** Deviations from the intended interventions **D3:** Missing outcome data **D4:** Measurement of the outcome **D5:** Selection of the reported result
	D1	D2	D3	D4	D5	Overall	
Barth [38] (2011)	(+)	(+)	(+)	(+)	(+)	(+)	
Liu (2009) [39]	(+)	(+)	(+)	(+)	(+)	(+)	
Adverse events related parameters							
Barth [38] (2011)	(+)	(−)	(+)	(−)	(+)	(−)	
Liu (2009) [39]	(+)	(−)	(+)	(−)	(!)	(−)	
Low risk (+)		Some concerns (!)		High risk (−)			

**Table 4 biomedicines-11-03180-t004:** Compiles the reported incidence of adverse events in the studies, highlighting various side effects associated with both interventions. Severe events, such as myocardial infarction, were observed in the PLEX group, while occurrences of hemolytic anemia during IVIG treatment were noted, albeit in a limited number of included patients.

Reported Side Effects in PLEX	Reported Side Effects in IVIG
hypertension (2) hematoma (1) citrate reaction (6) poor venous access delaying treatment (4) vasospasm (8) vasovagal reaction (2) myocardial infarction (1)	allergic reaction (2) nausea and vomiting (8) headache (8) chills (2) fever (3) hemolytic anemia (1) hypertension (1)

## Data Availability

All data generated or analyzed during this study are included in this published article and its Appendix A.

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
