# Peer review of "Plasma Exchange versus Intravenous Immunoglobulin in Worsening Myasthenia Gravis: A Systematic Review and Meta-Analysis with Special Attention to Faster Relapse Control"

_biomedicines, 2023, doi:10.3390/biomedicines11123180_

Round 1

Reviewer 1 Report

Comments and Suggestions for Authors

This metanalysis and systematic review article explores efficacy and safety of PLEX and IVIg in treatment of MG. Overall, the article is well-written. I find it very interesting, but I have some comments:

·      Introduction. Line 46. “pregnancy”. The role of pregnancy in MG is still debated. In most cases we observe amelioration of MG symptoms during pregnancy with a deterioration after delivery. Please be more precise.

·      Lines 55-56. I do not agree on times for initiation. This is not a problem. Both IVIg and PLEX can be started immediately. Probably more preparatory exams are needed to start PLEX…

·      Line 68. The problem with IVIg is not only the cost for production, but also the lack of volunteer blood donors as IVIg is a product from elaboration of plasma. Hence, there is low availability, also because IVIg are used for many other indications (CIDP, hematology, etc).

·      I suggest describing advantages and disadvantages of IVIg and PLEX in a Table to allow a simple comparison for the reader. A suggestion: the combination of PLEX and IVIg can be useful only if PLEX precedes IVIg. In this case a summation of effects can be achieved; on the contrary the effect of IVIg is overcome by elimination of all IgG through PLEX. It is a good motivation to do PLEX first in most severe cases. The problem with venous accesses in PLEX should be discussed.

·      Results. Are there any differences on treatment depending on serology? What about efficacy of PLEX and IVIg in AChR, MuSK or double-seronegative MG? Is there a difference in efficacy depending on serology or antibody titres? It may seem reasonable to think that PLEX could be more effective in the presence of high AChRAb titres. If it was not demonstrated, it should be specified.

·      The authors correctly conclude that these treatments are equal in efficacy, but I think that a reduction in 2.5 points in QMG is a great difference, even if still not significant. The low number of patients and studies limits the results. More studies are needed to explore the differences in efficacy. Another point is the quicker response in clinical practice after PLEX (a few days), when compared to IVIg (15-30 days). Unfortunately, these data are well-known by clinicians, but they are unpublished, as for the efficacy of steroids in MG.

·      I suggest to summarize adverse events in a small table. Were there any cardiac events or syncope, anemia after PLEX? And what about IVIg?

·      Among the limitations it should be stated clearly that the two studies examined in the metanalysis presented different dosage of IVIg and probably different number of PLEX sessions.

·      Future direction. I suggest to conclude this review with a paragraph on future direction discussing the potential of new studies comparing IVIg and PLEX and the role of new therapies as rescue therapy in MG: on this topic efgartigimod, a FcRn inhibitor which acts similarly to PLEX reducing total IgG (and pathogenetic antibodies) serum levels. As the mechanism is very similar to PLEX, efgartigimod acts very fast and early amelioration is achieved when compared to IVIg or other immunotherapies. Discuss its role citing recent evidence (Efgartigimod beyond myasthenia gravis: the role of FcRn-targeting therapies in stiff-person syndrome. J Neurol. 2023).

·      Style and grammar are fine. “QMGS” is not commonly used. Replace with the well-known acronym “QMG”.

Reviewer 2 Report

Comments and Suggestions for Authors

The authors present a manuscript in which a meta-analysis comparing plasmapheresis versus IVIG administration in patients with severe relapse of myasthenia gravis is performed.
They present only two studies since, in principle, no other studies meet the criteria.
The authors conclude in an adequate analysis that both methods have a similar behavior, although plasmapheresis acts faster, which could be interesting to avoid respiratory affectations.
In the abstract, plasma exchange initially appears with the initials PE and throughout the rest of the manuscript as PLEX.

Reviewer 3 Report

Comments and Suggestions for Authors

The authors analyzed the effects of PLEX or IVIG treatments during MG respiratory gravis and confirmed previously obtained results of equal efficacy and safety of both approaches. The authors found suitable for analysis only two studies and did not gain any additional information compared to previous reports.

Comments

1.      Abstract: The authors should disclose all the abbreviations and unify these abbreviations.

2.      Lines 30-32: It is not clear to which treatment this description is applied. This should be clarified.

3.      Lines 88-89: The rationale for this hypothesis should be presented.

4.      Line 110: “…AND random*” is not clear. This should be clarified.

5.      Line 124: “(16<)” is not clear. This should be clarified.

6.      Fig 1 is missing. This should be corrected.

7.      The legends for all Figures and Tables should be presented. This should be corrected.

8.      Discussion should not repeat Introduction (Lines 224-230; 263-266). All the repeated sentences should be removed (lines 274-275).

9.      Lines 259-262: It is not clear what are these sentences related to. This should be clarified.

10.  Line 266: The faster effect of PLEX should be numerically documented. This should be corrected.

11.  No references on Tables and Figures should be placed in Discussion and Conclusions sections. This should be corrected.

12.  Lines 283-289: These statements should be discussed.

Round 2

Reviewer 3 Report

Comments and Suggestions for Authors

I have no more comments.